# A Novel Baseline-Free Method for Damage Localization Using Guided Waves Based on Hyperbola Imaging Algorithm

**DOI:** 10.3390/s23042050

**Published:** 2023-02-11

**Authors:** Jichao Xu, Wujun Zhu, Xunlin Qiu, Yanxun Xiang

**Affiliations:** Key Laboratory of Safety Science of Pressurized System of MOE, School of Mechanical and Power Engineering, East China University of Science and Technology, Shanghai 200237, China

**Keywords:** Lamb waves, baseline-free, hyperbola, cross-correlation, damage localization

## Abstract

Most imaging methods based on ultrasonic Lamb waves in structural health monitoring requires reference signals, recorded in the intact state. This paper focuses on a novel baseline-free method for damage localization using Lamb waves based on a hyperbolic algorithm. This method employs a special array with a relatively small number of transducers and only one branch of the hyperbola. The novel symmetrical array was arranged on plate structures to eliminate the direct waves. The time difference between the received signals at symmetrical sensors was obtained from the damage-scattered waves. The sequence of time difference for constructing the hyperbolic trajectory was calculated by the cross-correlation method. Numerical simulation and experimental measurements were implemented on an aluminum plate with a through-thickness hole in the current state. The imaging results show that both the damages outside and inside the diamond-shaped arrays can be localized, and the positioning error reaches the maximum for the diamond-shaped array with the minimum size. The results indicate that the position of the through-hole in the aluminum plate can be identified and localized by the proposed baseline-free method.

## 1. Introduction

Plate-like structures are widely used in the fields of mechanical engineering, civil engineering, aerospace engineering and so on. During the fabrication and service process, plate-like structures inevitably undergo various degrees of damage with the possibility of failure under extreme conditions and environment. Structural health monitoring (SHM) based on Lamb waves is an important means to effectively monitor the structural safety and predict the service life of structures, due to its outstanding advantages of low attenuation, long propagation distance, large detection area and, most importantly, high sensitivity to structural damage and material inhomogeneity [1,2,3,4]. However, phenomena like dispersion, multimodes and the mode conversion (change in wave velocity when the wave interacts with structural discontinuities or boundaries) of Lamb waves make reliable damage localization a challenging task, because it is difficult to effectively extract damage features from the received signals. 

In the process of damage localization, the typical detection technology is to refer to the baseline signals, namely, to compare the signals in the current state with those in the healthy state. A scattered signal containing some damage-related information (location, size, orientation, type) can be obtained by performing reference signal subtraction. The time of flight (ToF), which corresponds to the time taken by the wave packet to travel from an actuator to a sensor along a certain path, is a straightforward feature of the damage-scattered signal for damage localization. The ellipse and hyperbola algorithms become the two most important methods for locating defects based on the ToF in plate-like structures [5,6,7,8,9,10].

This technology will be very effective and accurate if damage is the only factor causing the change in the baseline signals. In practice, operational and environmental conditions, such as stress, load, temperature, etc., are constantly changing, which may lead to significant changes in the baseline signals and can easily overwhelm the signal changes caused by damage [11,12]. This could cause significant errors and may even corrupt the damage localization results. 

To overcome such drawbacks, baseline-free damage detection algorithms have become a focus in the development of SHM. An instantaneous baseline can be obtained through a comparison between different sensor pairs that have a similar path [13], but the distance of each pair must be kept identical. A large number of undamaged wave-propagating paths are also required, which may not be satisfied simultaneously in practice. The time reversal method (TRM) is widely used as a promising candidate for baseline-free damage detection in thin-walled plate/shell-like structures [14]. It is based on the concept of spatial reciprocity and time-reversal invariance, originally observed in acoustic waves [15]. This is due to an input signal, which can be reconstructed at the source transducer, when the output signal measured at the receiver transducer is time-reversed and emitted back. Liu et al. [16] combined the probability imaging algorithm with the virtual time reversal method to study the damage in a composite board by air-coupled sensors; the effects of the damage type and size on the probability imaging algorithm were discussed to achieve the baseline-free detection. Huang et al. [17] proposed a reciprocity index-based probabilistic imaging algorithm to achieve the baseline-free identification of delamination on the path of a piezoelectric piece. Kannusamy et al. [18] presented a refined time reversal method in conjunction with the probability imaging algorithm for baseline-free damage localization in thin plates, using a sparse array arrangement of piezoelectric wafer transducers.

However, the aforementioned methods need a lot of actuators/sensors to form enough monitoring paths for damage localization. Damage in the survey areas, but not covered by the monitoring paths, is hard to detect accurately. To circumvent these limitations, damage can be detected and localized anywhere in the monitoring areas by using the ToF of the damage-scattered signals. Wang and Yuan [19] applied the time reversal method on the scattered signals so as to focus on the damage location and achieve the baseline-free damage imaging using a delay and sum algorithm. Jeong et al. [20] obtained the ToF information using the time reversal of a scattered Lamb wave and constructed a baseline-free localization image by a beamforming technique. Jun and Lee [21] proposed a new baseline-free technique, which combined a hybrid time reversal method with a beamforming method using the ToF of the wave packets generated by the damage. In general, TRM often requires a programmable transducer array to synthesize time-reverse signals simultaneously, which limits its application due to the complexity and cost of the hardware system. Therefore, a baseline-free method without complex manipulation, such as time-reversal based on the ToF information, is urgently needed.

In addition to the ToF of the scattered signals, the time difference of arrival (TDOA) between two scattered signals can also be used for damage localization. This method utilizes the difference in the ToF to localize damage based on the intersection of hyperbolas. Yelve et al. [22] introduced a baseline-free method based upon the TDOA of nonlinear Lamb waves to localize barely visible impact damage in composite laminates. Li et al. [23] introduced a probability-based hyperbola diagnostic imaging method based on the different ToFs of the damage signals to achieve the baseline-free damage localization and imaging. This method required at least twice as many transducers as in the general hyperbola method to form sufficient pairs of two sensors. This study aims to achieve baseline-free damage localization by the hyperbola algorithm using far fewer transducers. A new array composed of these transducers can not only eliminate the direct wave, but can also make the boundary reflection signal, and other noise, cancel each other out. The proposed baseline-free method is expected to be more convenient and effective in engineering practice.

According to the aforementioned research, a crucial step of the baseline-free damage imaging, based on the ToF, is to extract the damage-scattered signals accurately. In this paper, a novel baseline-free method for damage localization was presented, including a peculiar symmetrical array of PZT actuators/sensors to eliminate the direct waves from the received signals. Meanwhile, the cross-correlation method (CCM) was introduced to determine the sequence of time difference accurately from the damage-scattered waves. The damage was identified and localized by a hyperbolic algorithm with only one branch of the hyperbolic curve. To demonstrate the feasibility and the effectiveness of the proposed strategy, numerical simulation and experimental validation were carried out to localize a through-thickness hole in an aluminum plate. The influence of different damage locations and array forms on the damage localization results was discussed.

## 2. Methodology for Baseline-Free Damage Localization

Lamb wave based damage localization is based on the fundamental idea that a traveling wave in a plate structure will be scattered by the damage. Note that the damage location is usually determined by the group velocity and the ToF of the scattered Lamb waves. While the group velocity of a certain Lamb wave mode is often determined in a homogenous and isotropic plate structure, the ToF information becomes a key factor in damage localization. There are many imaging and localizing methods based on the ToF, including ellipse and hyperbola algorithms. 

Whereas the ellipse algorithm considers two transducers to be acting in pairs, the hyperbola algorithm mainly considers combinations of one actuator (transmitter), n, and pairs of sensors (receivers), i and j, in a spatially distributed array, as shown in Figure 1a. Without the loss of generality, a received signal on the sensing path consists, at least, of a direct signal and a scattered signal. If a damage is positioned at the point D(x,y), the difference in arrival time of the scattered signals at the two sensors is defined as:(1)$$\Delta t_{ij}\left(x,y\right)=\frac{l_{n-D-i}-l_{n-D-j}}{V_{g}},$$
where $l_{n-D-i}$ and $l_{n-D-j}$ are the distances from the actuator to the damage, and then to the two sensors, respectively. Further, $V_{g}$ is the velocity of the signals. Since the distance from the actuator to the damage is the same for both sensors (Figure 1a), it does not affect the difference in the arrival time. Equation (1) can then be simplified as [24]: (2)$$\Delta t_{ij}\left(x,y\right)=\frac{\sqrt{{\left(x_{i}-x\right)}^{2}+{\left(y_{i}-y\right)}^{2}}-\sqrt{{\left(x_{j}-x\right)}^{2}+{\left(y_{j}-y\right)}^{2}}}{V_{g}},$$
where $\left(x_{i},y_{i}\right)$ and $\left(x_{j},y_{j}\right)$ are the coordinates of the two sensors, respectively. The solution of Equation (2) is a hyperbola with its foci on the two sensors. By performing this for all the available nij groups of sensors, the damage location in the sensor network is obtained as the intersection point of all the hyperbolas. The time difference of the scattered signals is a robust feature of the wave signals, for which it is not necessary to know the time origin of the excitation. 

In general, the scattered signals are always subtracted by the corresponding baselines in the intact state, depicted by:(3)$$f_{SC}^{i}\left(t\right)=f_{S}^{i}\left(t\right)-f_{C}^{i}\left(t\right),$$
where $f_{SC}^{i}\left(t\right)$ is the scattered signal for the sensor i, $f_{S}^{i}\left(t\right)$ is the received signal for the sensor i in the intact state, which is the state without damage in the structure, and $f_{C}^{i}\left(t\right)$ is the received signal for the sensor i in the current state. The same processing is carried out for the sensor j. To realize baseline-free localization and imaging, a novel symmetrical sensor arrangement is presented to eliminate the direct propagating waves and extract the damage-scattered waves in this paper. As shown in Figure 1b, one actuator is placed in the perpendicular bisector of the line segment connecting the two sensors, while one actuator and two sensors are located in random positions in the typical sensor array. This form of array can be in a diamond or square shape, as long as the diagonals of the array are perpendicular to each other. Since the distances between the actuator and the two sensors are equal, according to vertical theorem in mathematics, the direct propagating signals on the two sensors from the actuator are almost identical. Under this situation, the damage-scattered signals can be extracted by subtracting the two received signals in the current state, which is described by: (4)$$f_{ij}\left(t\right)=f_{i}\left(t\right)-f_{j}\left(t\right),$$
where $f_{ij}\left(t\right)$ is the damage-scattered signal; $f_{i}\left(t\right)$ and $f_{j}\left(t\right)$ are the received signals for the sensors i and j in the current state, respectively. At the same time, boundary reflection signals, and other noise, can also cancel each other out because of the symmetrical positioning of the two sensors in the structure. Therefore, the time difference $\Delta t_{ij}\left(x,y\right)$ can be obtained according to the characteristics of the damage-scattered signals. Here, Hilbert transform [12] is used to estimate the ToF by extracting the time corresponding to the maximum of the envelope of the signal component. This is described by:(5)$$H_{ij}\left(T\right)=\frac{1}{\pi }\overset{+\infty }{\underset{-\infty }{\int }}\frac{f_{ij}\left(t\right)}{T-t}dt,$$
where $H_{ij}\left(T\right)$ is the Hilbert transform of the damage-scattered signal $f_{ij}\left(t\right)$.

In addition, one should also pay further attention to the sign of the time difference $\Delta t_{ij}\left(x,y\right)$, because the hyperbolic loci usually have two branches by definition, which may bring pseudo-intersection points. To overcome such a problem, a cross-correlation method (CCM) is used to determine the sequence of time difference with a positive or negative sign. The time delays are calculated by means of the CCM between the damage-scattered signals and the two received signals, respectively, in which the cross-correlation function exhibits a notable characteristic value at $\Delta t_{ij}\left(x,y\right)$. This can be expressed by [25]:(6)$$R_{iji}\left(T\right)=\overset{+\infty }{\underset{-\infty }{\int }}f_{ij}\left(t\right)f_{i}\left(t-T\right)dt=\overset{+\infty }{\underset{-\infty }{\int }}f_{ij}\left(t+T\right)f_{i}\left(t\right)dt$$
and
(7)$$R_{ijj}\left(T\right)=\overset{+\infty }{\underset{-\infty }{\int }}f_{ij}\left(t\right)f_{j}\left(t-T\right)dt=\overset{+\infty }{\underset{-\infty }{\int }}f_{ij}\left(t+T\right)f_{j}\left(t\right)dt.$$

Since the damage-scattered signals are subtracted from the two received signals, the cross-correlation function will give a peak value at time 0, simultaneously. If the peak value appears at $\Delta t_{ij}\left(x,y\right)$ positive relative to 0, the scattered signal in another received signal will be behind that in the received signal. Otherwise, it is in advance. Once the sign of $\Delta t_{ij}\left(x,y\right)$ has been determined, whether it is positive or negative, it can specify only one branch of the hyperbola, so as to achieve damage localization precisely.

In this imaging method, a cumulative distribution function (CDF) $F\left(z_{ij}\right)$ is introduced to determine a specific pixel at each node when the inspection area of the structure is virtually and evenly meshed, which is defined as [26]: (8)$$F\left(z_{ij}\right)=\int_{-\infty }^{z_{ij}}f\left(z\right)dz.$$
where $f\left(z\right)=\frac{1}{\sigma \sqrt{2\pi }}\exp\left[-\frac{z}{2\sigma }\right]$ is the Gaussian distribution function that relates the probability density of damage occurrence at the meshed node (x,y). Further, σ is the standard deviation of the relevant damage feature as a tolerance factor in the imaging process, and $z_{ij}$ is the upper limit of integral function, which means the shortest distance from the node to the hyperbolic loci. Thus, the probability value $D_{ij}\left(x,y\right)$ on each node constructed by sensor i and j is then defined as:(9)$$D_{ij}\left(x,y\right)=1-\left[F\left(z_{ij}\right)-F\left(-z_{ij}\right)\right].$$

For simplicity, a general overview of the proposed method in the symmetrical array is given in Figure 2. 

## 3. Validation of Numerical Simulation

In order to verify the effectiveness of the proposed method, a three-dimensional finite element (FE) model was firstly developed in conjunction with a commercial FEM software (ABAQUS/CAE). The FE model was considered as an effective method to study the characteristics of wave propagation in plate structures. 

### 3.1. Simulation Model

The dimensions of the aluminum plate employed for the simulation were 300 mm × 300 mm × 1 mm. A circular through-thickness hole with a diameter of 5 mm was dug on the aluminum plate as a damage source, as shown in Figure 3. Additionally, in order to realize very minimal or no wave reflection from the boundary, a gradually damping artificial boundary was introduced to simulate a nonreflecting boundary condition [27]. The damping coefficients of the individual layers were gradually increased from the inner layer to the finite boundary. Both the boundary length and the damping coefficient of each layer were dependent on the excitation frequency. The Young’s modulus, Poisson’s ratio and the density of the aluminum plate were 69 GPa, 0.33 and 2700 kg/m^3^, respectively.

Since piezoelectric elements were not available in ABAQUS/explicit, concentrated forces were applied to simulate the Lame wave excitation for round PZT transducers. Hanning window modulated five-cycle sinusoidal tone bursts, at a central frequency of 250 kHz, were excited by applying a pair of symmetric forces on two points, which were symmetrically positioned on the upper and lower surfaces of the plate. This kind of point displacement modeling to generate a single S0 mode Lamb wave was described in detail in [28]. The S0 mode Lamb wave was selected as an incident wave because it is more sensitive to the through-hole than an A0 mode. The Lamb waves were received at monitored points on the upper surface of the plate. 

A global cartesian coordinate was introduced, in which the abscissa axis was parallel to the lower edge of the plate, and the bottom left corner of the plate was at the origin. The points for exciting and receiving signals, forming a diamond-shaped array, were labelled as A, B, C and D at different coordinates, respectively. The coordinates of the diamond-shaped array, and different damage positions, were shown in Table 1; the coordinates of the diamond-shaped arrays with different sizes and the damage location, were shown in Table 2. These diamond-shaped arrays were numbered from 1 to 6, both in the case of numerical simulation and experimental testing. 

Eight-noded 3D reduced integrated linear brick elements, C3D8R, were adopted to complete the structured mesh division. In order to obtain sufficient accuracy and high efficiency, the dimension of the elements was around 0.5 mm × 0.5 mm × 0.5 mm, so that there were at least 20 elements per wavelength of the Lamb waves. A time step of 20 ns was used to ensure the accuracy of the wave propagation modeling [29]. 

### 3.2. Signal Analysis

According to the above-mentioned diamond-shaped array for excitation and reception, all the received signals were obtained from the pair of two sensors. As shown in Figure 4, there are two time-domain received signals from the sensing path AB (blue line) and AD (red line) in the diamond-shaped array with number 1, including a direct signal and a scattered signal, respectively. The direct signals are almost the same in the two received signals. By taking the difference between the two received signals (Equation (4)), a damage-scattered signal was obtained by eliminating the direct signals from each other. 

As shown in Figure 5a, the two wave packets in the damage-scattered signal represent the scattered signals in the received signals, respectively. Since the direct propagation distances from one actuator to pairs of two sensors are equal theoretically, the direct signals are canceled by each other. Moreover, since the two receiving sensors are geometrically symmetrical in the structure, other noise and boundary reflection signals are also eliminated in the subtraction process, so as to highlight the damage-scattered signals. The envelope curve (blue line in Figure 5a) was constructed by Hilbert transform (Equation (5)) from the damage-scattered signal, hence the relative time difference Δt was determined from the corresponding times (two red dots of the envelope curve in Figure 5a). According to the Δt and group velocity of the S0 mode, the hyperbolic trajectory is constructed, and the points on the trajectory indicate the possible locations of the damage. Obviously, the group velocity of the S0 mode can be calculated from the received signals with different propagation distances, which is about 5164.7 m/s in this numerical simulation. 

At the same time, the sequence of Δt was determined by means of the CCM (Equations (6) and (7)) between the damage-scattered signal and the two received signals, respectively. As shown in Figure 5b, there are two envelope curves based on the cross-correlation function, in which a peak always appears at time zero. Considering time zero as the reference, and the relative time difference Δt as the interval, the corresponding peaks at Δt are found on the left- or right hand sides of the time zero, respectively. Therefore, the sign of Δt is determined when the time sequence is judged from the time lag in the cross-correlation curve. For instance, the red envelope curve was calculated from the cross-correlation between the damage-scattered signal and the received signal AD. Obviously, the peak at Δt is on the left hand side of the time zero, which means that the scattered signal in the received signal AB is in advance of that in the received signal AD. On the contrary, it is behind. Similar results can be obtained from the blue envelope curve. Unfortunately, the peak may not be easy to find in the blue envelope curve because of the interference of the other components of the cross-correlation function. Nevertheless, any of the envelope curves can be used to determine the sequence of the time difference Δt, and there is no need to calculate the value of the relative time difference. This may be a relatively robust feature for damage localization. 

After obtaining the time difference, the damage location can be realized based on the hyperbolic trajectory; the imaging results of the through-hole location in the simulation is shown in Figure 6a. The maximum probability position of the imaging result is at 60.1, 240.1, while the center position of the through-hole is at 50, 250. It is shown that the proposed method can be used to achieve baseline-free damage localization. At the same time, the imaging result of the traditional hyperbolic method in the symmetrical array is shown in Figure 6b. The coordinates of the maximum pixel value are 56.5, 243.5. It shows that the traditional hyperbolic method can also achieve damage positioning, and is more accurate than the proposed method. The reason is probably due to the use of more groups of signals in the traditional hyperbolic method; there are 12 pairs of signals in total, while the proposed method includes four groups of signals. It should be noted that the more signals, the longer the processing time of localization, and the traditional method needs the reference signals in the intact state. 

## 4. Experimental Verification

The proposed method based on Lamb waves was then corroborated by the ultrasonic experimental testing of a through-hole in an aluminum plate. 

### 4.1. Experimental Setup

The experimental testing system, as shown in Figure 7, consisted of a dual channel arbitrary/function generator (Textronix AFG 3022C), a power amplifier (EPA 104), a mixed domain oscilloscope (Textronix MDO 3012) and a personal computer (PC) connected with the oscilloscope. The specimen was a common aluminum plate with a dimension of 500 mm × 500 mm ×1 mm, and a through-hole with a diameter of 5 mm was used as a damage source, which was the same as that in the numerical simulations. Four PZT piezoelectric transducers, each 10 mm in diameter and 1 mm in thickness, were permanently bonded to the surface of the aluminum plate by means of a thin cyanoacrylate adhesive layer. 

A global cartesian coordinate was also introduced for convenience, as in the simulation model. The PZT transducers were labelled as A, B, C and D; damage locations also had the same coordinates as in the numerical simulation. It should be noted that the bottom left corner of the aluminum plate was not at the origin of the coordinate axis. 

A five-cycle Hanning windowed sine wave with a central frequency of 250 kHz, generated by the arbitrary function generator, was applied as the excitation tone burst. The burst was then amplified by means of the power amplifier before it was sent to one of the PZT transducers. The PZT transducers, at two symmetrical positions, were used to receive the Lamb wave signals. For example, the PZT transducers B and D, which were considered to be symmetrical in the diamond-shaped array, were used to receive the signals when PZT transducer A was excited. The signals were captured by the oscilloscope at a sampling rate of 25 MHz with 256-time averaging, and was then transmitted to the computer. The excitation-acquisition procedure was implemented in turn on each PZT transducer, so that in total eight received signals were collected from these sensing paths. It is worth noting that the received signals were always measured in the current state, but not in the intact state. 

### 4.2. Signal Processing

The experiments and simulations have some similar results because of the same excitation and reception process with the diamond-shaped array. Firstly, the group velocity of the S0 mode is about 5451.2 m/s, as calculated from the direct waves with different propagation distances. It is considered that the PZT transducers can effectively excite the S0 mode Lamb wave. Then, the received time-domain signals of the sensing paths AB and AD, in the diamond-shaped array with number 1, are represented in Figure 8, including the direct waves and the scattered waves. Meanwhile, there is an electromagnetic interference signal at the front of the received signals, which is mainly introduced by the transducer and circuit imperfections. 

After the electromagnetic signal was filtered out, the two received signals were compared, according to Equation (4). A damage-scattered wave was obtained by eliminating the direct waves via subtraction of the received signals. However, as shown in Figure 9a, the damage-scattered wave still has an obvious wave packet at the position of the direct wave, which means that the direct waves are not fully consistent with each other and not completely eliminated, as in the numerical simulation. This is because the direct waves are not exactly the same in phase and amplitude, as shown in the inset of Figure 8. This is due to a slight difference in the performance of the adhesive and the geometric position of the PZT transducers, which cannot be avoided in the experiments. Fortunately, the difference between the two direct waves can be reduced considerably by phase compensation and standard normalization to obtain the damage-scattered wave, as shown in Figure 9b. The next steps are consistent with those in the simulation, including calculating the time difference Δt and constructing the hyperbolic trajectory. It is noted here that the first wave packet in the damage-scattered wave (the first red dot shown in Figure 9b) may be the A0 mode direct wave. This is because the A0 mode could also be excited in the above process of the effective S0 mode excitation, in which the amplitude of the A0 mode is seemingly small. Although the S0 mode direct wave can be eliminated through the above manipulation, the A0 mode direct wave may not be completely excluded. 

At the same time, the sequence of Δt was determined by means of the CCM (Equations (6) and (7)). As shown in Figure 10, the red envelope curve is the cross-correlation curve between the damage-scattered wave and the received signal AD. A characteristic point corresponding to the time difference Δt is found on the left hand side of the time zero, shown as a black circle in Figure 10. It indicates that the scattered wave in the received signal AB is in advance of that in the received signal AD. On the other hand, the blue envelope is the cross-correlation curve between the damage-scattered wave and the received signal AB, and it does not show the characteristic point at the time difference Δt. Regardless, the time sequence could be obtained as long as a feature point in any of the cross-correlation curves is found. It should be noted that the peak of the cross-correlation curve does not appear at the time zero. This is because there is a time offset in the process of phase compensation. In addition, the point corresponding to the time difference Δt is actually in the valley of the signals, not at the peak. This could be attributed to the fact that the first red dot corresponding to the time difference Δt of the damage-scattered signal is also in the trough of the waves, which is caused by the failure to completely eliminate the A0 mode direct wave. 

## 5. Results and Discussion

In the aforementioned diamond-shaped array, the time difference Δt is obtained from the two received signals in the current state. Then, using the proposed baseline-free method, the imaging results of damage locations in the experiments are shown in Figure 11 and Figure 12. 

As shown in Figure 11a, there are four intersections of the hyperbolic trajectories in the diamond-shaped array, resulting in artifacts in the imaging result, and it is not able to achieve damage localization accurately. Obviously, the hyperbola itself has two branches using the relative time difference of each pair of the two received signals. Therefore, it is still necessary to determine the sign of the relative time difference (namely, the time sequence), for precise damage localization. By using the cross-correlation function, only one branch of the hyperbola can be determined and the damage location can be achieved precisely, as shown in Figure 11b. The maximum probability position of the imaging result considered to be the most likely location for the damage is at 35.6, 266, while the center position of the through-hole is at 50, 250. According to the error calculation method in [30], the relative error of the imaging result is about 5.3%. It shows that the probability position of the imaging result is very close to the actual location of the through-hole, and the method gives the damage location fairly well. 

The imaging result in Figure 12 is similar to that in Figure 11, while the most obvious difference is that in Figure 12 the damage location is inside the diamond-shaped array and on the perpendicular bisector of certain line segment(s) of the pairs of two receiving sensors. Here, a correlation coefficient of two received signals is used to prejudge whether the damage location is on some of the perpendicular bisectors. If the correlation coefficient exceeds a certain threshold, it means that the damage is most likely on the perpendicular bisector, and vice versa. As shown in Figure 12b, the maximum probability position of the imaging result is at 92.9, 149.8, which is almost at the actual center position of the through-hole at 100, 150, with a relative error of only about 2.36%. The above results show that no matter whether the damage location is inside or outside the diamond-shaped array, or even in the position symmetrical with respect to the pairs of two receiving sensors, the proposed method can be used to achieve baseline-free damage localization. 

In order to further study the influence of the array forms on the imaging results, diamond-shaped arrays with different sizes were used for the baseline-free damage localization of the aluminum plate. As shown in Figure 13, the imaging results in the experiments are obtained for different arrays. Meanwhile, the size of the arrays, the damage location and relative errors are shown in Table 2. As shown in Figure 13a, when the diagonal distance of the diamond-shaped array is about 80 mm, the maximum relative error is 7.7%. Compared with the array with maximum size, the array with minimum size has not only a higher relative error, but also two maximum probability locations in the simulated imaging results. It is worth mentioning that these imaging results come from some intersecting areas, not trajectory intersections; this is because this size of array has a small time difference, which makes the hyperbolic trajectory infinitely approach their asymptotes. Consequently, the maximum probability position becomes the intersection area of the asymptotes. Based on the above research, it is believed that when the diagonal distance of the diamond-shaped array is less than 80 mm, it will deteriorate the positioning accuracy and cause noticeable positioning errors, no matter where the damage location is. However, as the size of the array (namely, the diagonal distance) increases, the maximum probability position approaches the actual locations of damage and the accuracy of damage localization also increases correspondingly. As shown in Figure 13d, when the diagonal distance of the diamond-shaped array is about 200 mm, the maximum probability location of the imaging result is at 241.4, 239.3, while the actual center position of the through-hole is at 230, 230. The relative error is about 3.8%, which means that the imaging result is close enough to the damage location. Therefore, the positioning accuracy can be significantly improved when the diagonal distance is larger than 80 mm, especially up to 200 mm. In summary, by using the diamond-shaped arrays with appropriate size, and the hyperbola algorithm, it is possible to effectively achieve damage localization with the proposed baseline-free method. 

The simulations and experiments have similar imaging results because of the same excitation and reception process in the diamond-shaped arrays. The positioning errors of arrays with different sizes in the experimental testing and numerical simulations are shown in Table 2. The results show that the accuracy of damage localization in the simulations is higher than that in the experiments. The reason may be that it is easy to excite a single S0 mode in the simulations, and the wave propagation is relatively simple, while the A0 mode can also be excited in the experiments, and the imaging results are probably affected by the multimode characteristics of Lamb waves. It can be seen that the damage localization is realized through simulations and experiments, which indicates that the proposed method is an effective baseline-free method for damage localization. 

In further study, in order to verify the effectiveness of the proposed method, a comparative analysis was made of the baseline-free method presented in Ref. [31]. The array form in the baseline-free method is shown in Figure 14a. Although it is somewhat similar to the array form mentioned above, the difference is that two transducers are closely placed as a pair in the special transducer arrangement. Thus, the direct wave and damage-scattered wave can be separated effectively because of the large difference of their time of flight. The damage-scattered wave is easily extracted, and the imaging result of through-hole location in the experiment is shown in Figure 14b. The maximum probability position of the imaging result is at 94.3, 146.3; it shows that the baseline-free method can also achieve damage positioning. Compared with the coordinate 92.9, 149.8 of the maximum value in Figure 12b, it could be concluded that the positioning accuracy of the two methods is almost the same, even though it seems that the baseline-free method is more accurate than the proposed method, according to the positioning error here. Thus, these two kinds of baseline-free method can achieve damage localization by the special arrangement. It is worth noting that there are four groups of signals in these methods, while only four transducers are used in the proposed method. Further, the baseline-free method in Ref. [31] has a high resolution and can detect multi defects due to the employed single mode nondispersive SH0 wave. 

## 6. Conclusions

Due to the changing of operational and environmental conditions, typical detection methods involving baseline signals have large damage localization error. To address this problem, a novel baseline-free method for damage localization using Lamb waves based on a hyperbolic algorithm, was proposed in this study. 

To eliminate the direct signals, diamond-shaped arrays were arranged on the plate structures to receive signals at two symmetrical positions. The time difference between the received signals was obtained from the damage-scattered signals. The sequence of time difference calculated by the cross-correlation method determined only one branch of the hyperbola. Damage localization was realized by a hyperbolic algorithm with the damage probability of each point in the structure. Numerical simulations and experimental measurements were implemented on an aluminum plate with a through-thickness hole in the current state. The imaging results show that the damage localization can be achieved no matter whether the position of the defect is outside or inside the diamond-shaped arrays. Meanwhile, because of the intersection of the asymptote of the hyperbola, the diamond-shaped array with the smallest size has the biggest positioning error of about 7.7%. Nevertheless, the relative error can be reduced to 3.8% when the diagonal distance of the diamond-shaped array increases to 200 mm. The consistency between the numerical and experimental results indicates that the position of the through-hole in the aluminum plate can be accurately identified and localized by the proposed baseline-free method. Ultimately, the baseline-free method proposed here may largely improve the practicability of the Lamb wave damage localization technique.

## Figures and Tables

**Figure 1 sensors-23-02050-f001:**
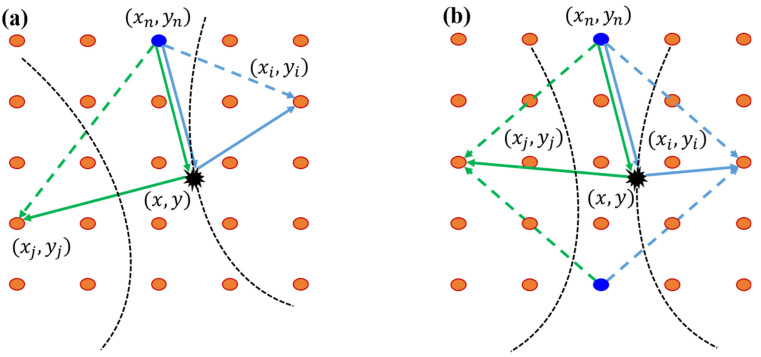
Schematic diagram of hyperbolic positioning with: (**a**) a traditional array; and (**b**) a new symmetrical array for exciting and receiving signals.

**Figure 2 sensors-23-02050-f002:**
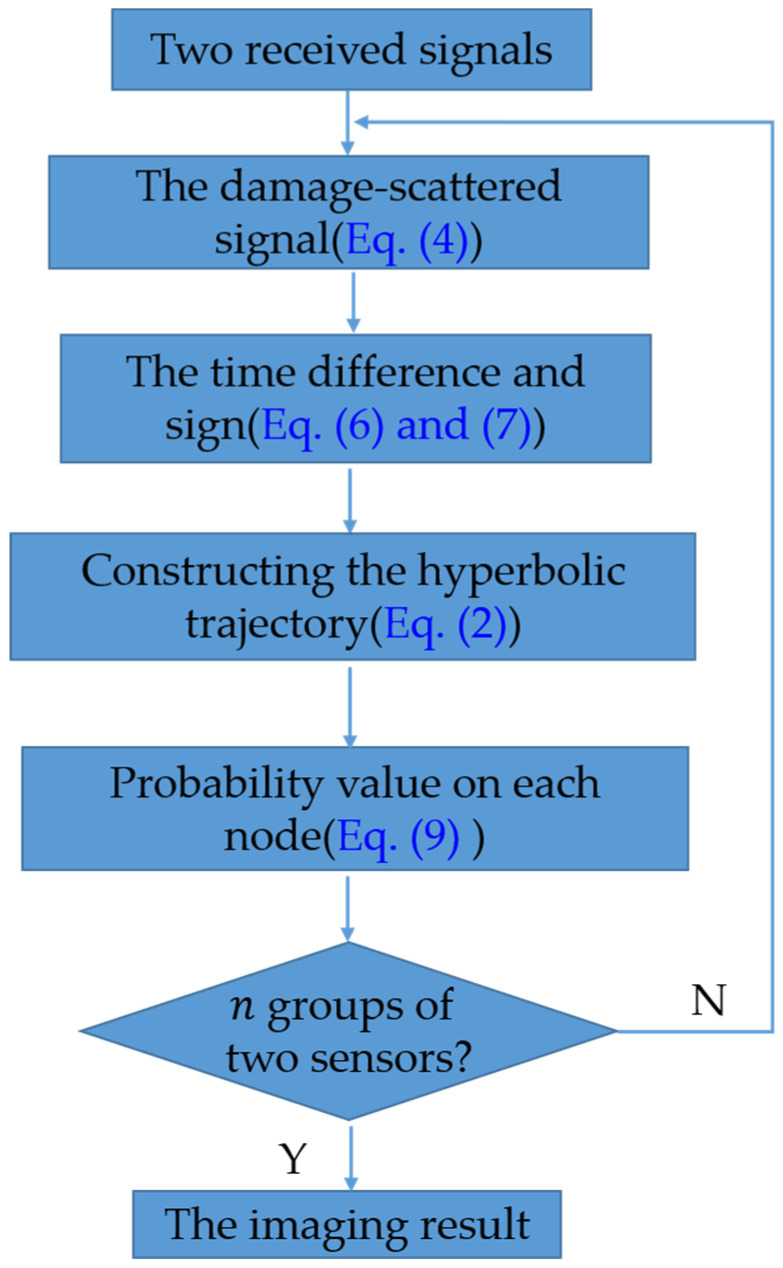
A flowchart of the proposed method in the symmetrical array.

**Figure 3 sensors-23-02050-f003:**
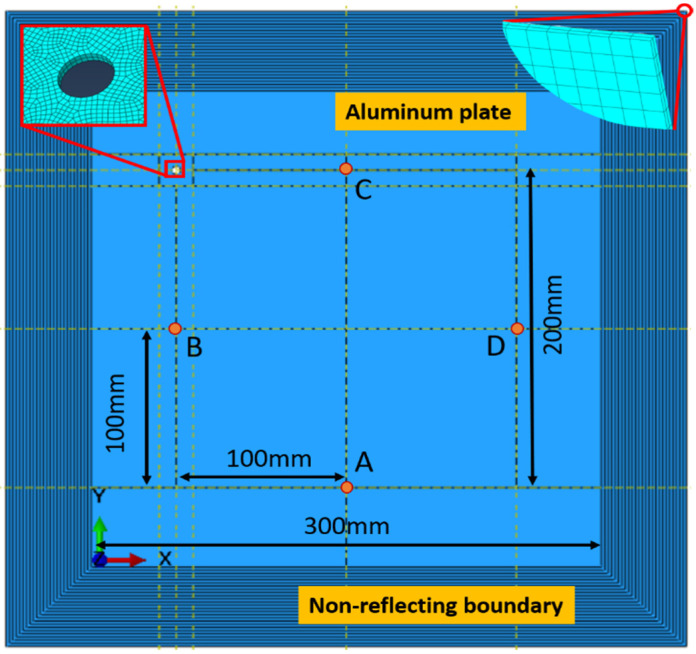
A schematic diagram of the simulation model setup.

**Figure 4 sensors-23-02050-f004:**
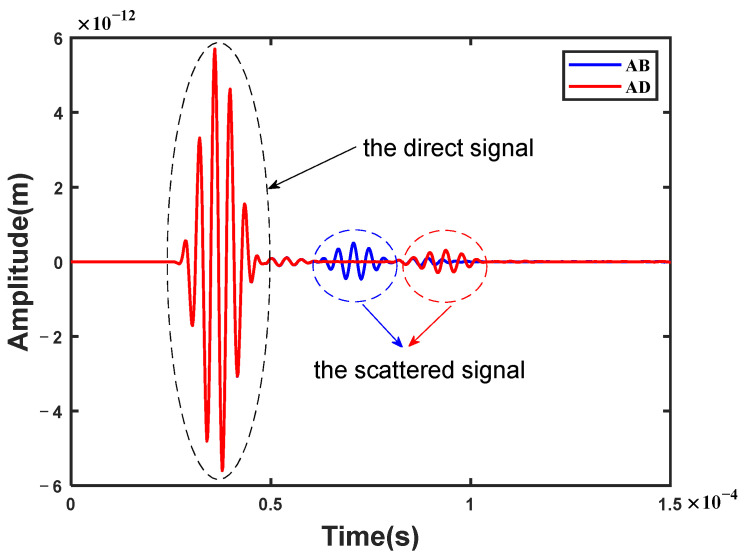
A schematic diagram of two received signals from the sensing path AB (blue line) and AD (red line) in the diamond-shaped array with the damage location of number 1.

**Figure 5 sensors-23-02050-f005:**
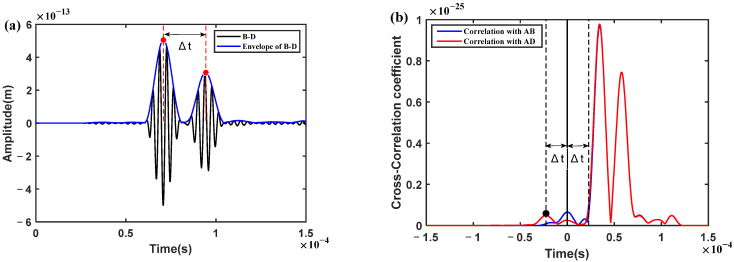
(**a**) A damage-scattered signal obtained by subtracting two received signals; (**b**) the cross-correlation curves between the damage-scattered signal and the two received signals, respectively. (Red dots denote the amplitude corresponding to the time, black dot denotes the peak corresponding to the time difference).

**Figure 6 sensors-23-02050-f006:**
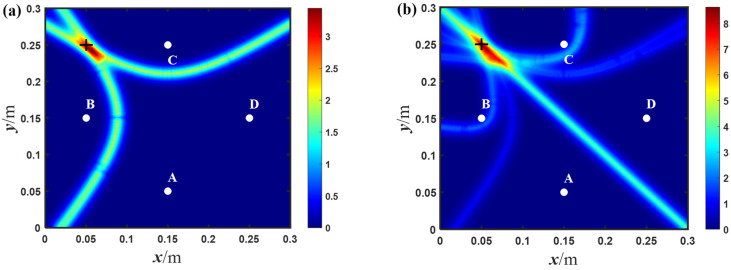
The imaging results of: (**a**) the proposed method; and (**b**) the traditional hyperbolic method in the diamond-shaped array with number 1 in the simulation (white points denotes the exciting and receiving position, + indicates the center position of the circular hole).

**Figure 7 sensors-23-02050-f007:**
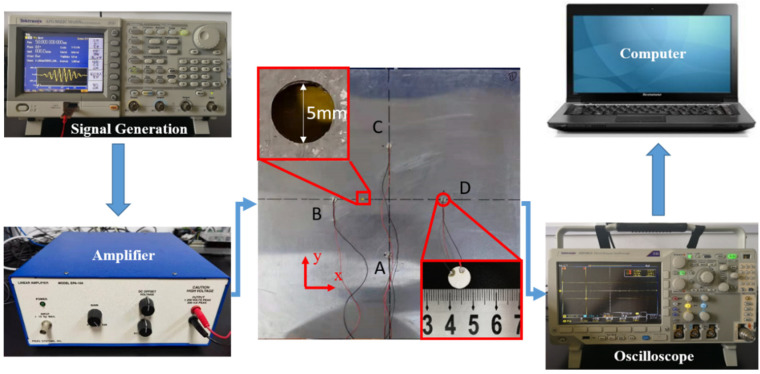
A schematic of the experimental testing system.

**Figure 8 sensors-23-02050-f008:**
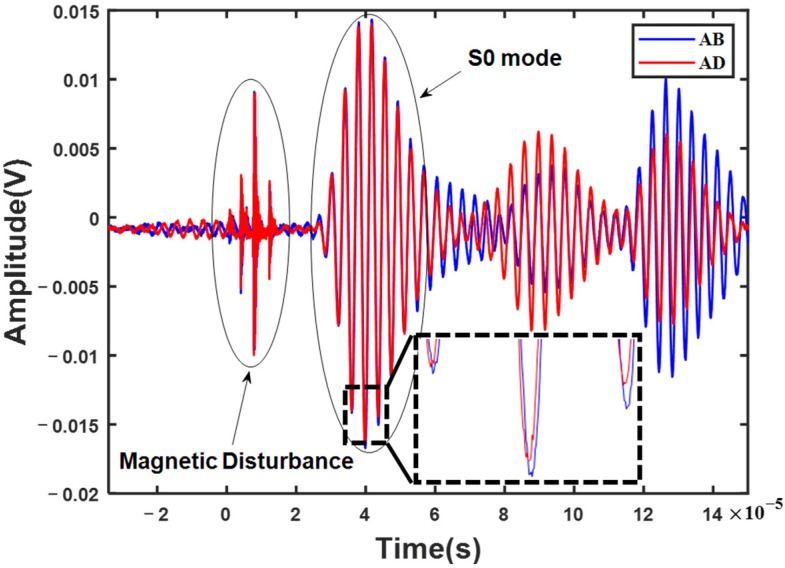
A schematic diagram of two received signals from the sensing path AB (blue line) and AD (red line) in the diamond-shaped array with the damage location of number 1.

**Figure 9 sensors-23-02050-f009:**
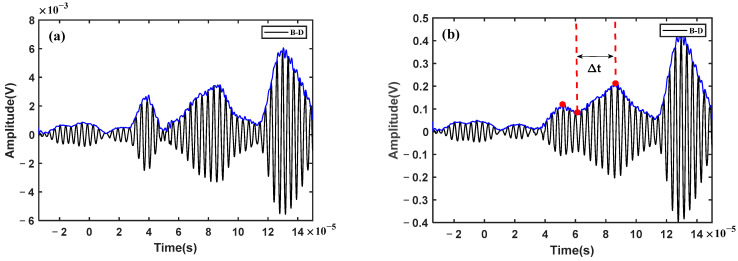
A damage-scattered signal obtained by subtracting two received signals: (**a**) without phase compensation; and (**b**) with phase compensation. (Red dots denote the amplitude corresponding to the time).

**Figure 10 sensors-23-02050-f010:**
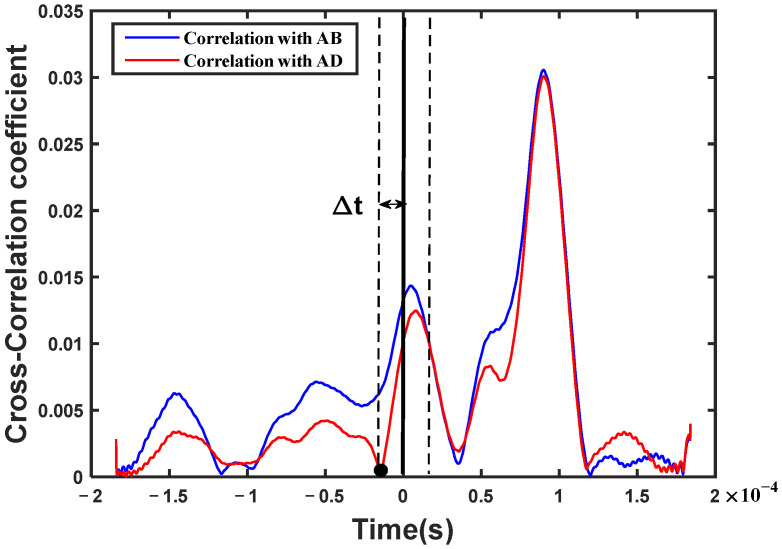
The cross-correlation curves between the damage-scattered signal and the two received signals, respectively. (Black dot denotes the characteristic point corresponding to the time difference).

**Figure 11 sensors-23-02050-f011:**
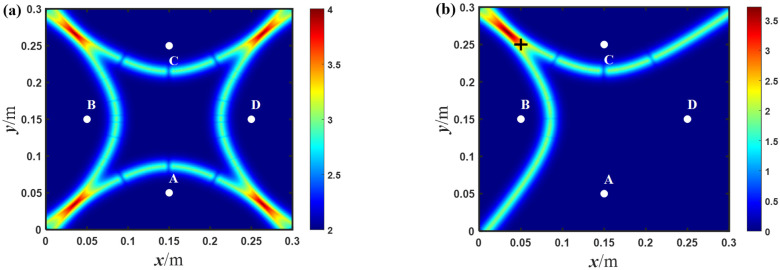
The imaging results of the experiments with: (**a**) two branches; and (**b**) one branch of hyperbola in the diamond-shaped array with number 1 (white points denotes the exciting and receiving position, + indicates the center position of the circular hole).

**Figure 12 sensors-23-02050-f012:**
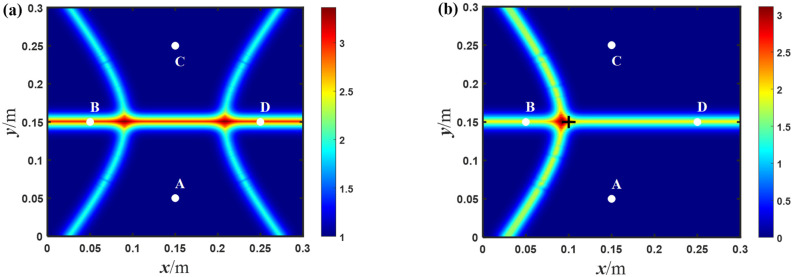
The imaging results of the experiments with: (**a**) two branches; and (**b**) one branch of hyperbola in the diamond-shaped array with number 2 (white points denotes the exciting and receiving position, + indicates the center position of the circular hole).

**Figure 13 sensors-23-02050-f013:**
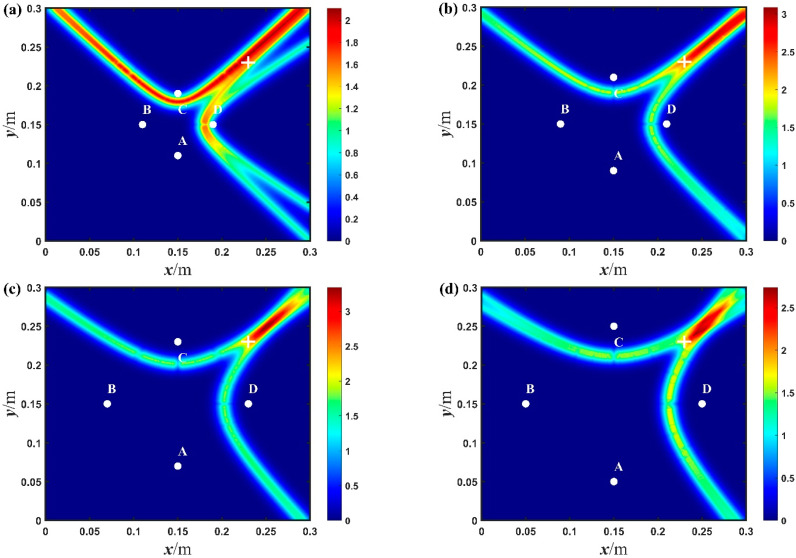
The imaging results of the through-hole for different diamond-shaped arrays with a diagonal distance of: (**a**) 80 mm; (**b**) 120 mm; (**c**) 160 mm; and (**d**) 200 mm, in the experiments (white points denotes the exciting and receiving position, + indicates the center position of the circular hole).

**Figure 14 sensors-23-02050-f014:**
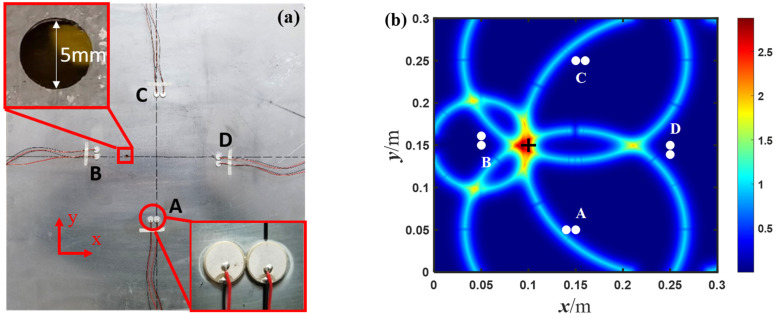
(**a**) A special transducer arrangement and (**b**) the imaging result of the baseline-free method [31] in the experiment (white points denotes the exciting and receiving position, + indicates the center position of the circular hole).

**Table 1 sensors-23-02050-t001:** The diamond-shaped array with the through-hole of different positions. (Unit: mm).

Position of Array	Center Position of the Hole	Localization Result (Simulation)	Error	Localization Result (Experiment)	Error	Number
Diagonal distance was 200, A (150, 50), B (50, 150), C (150, 250), D (250, 150).	(50, 250)	(60.1, 240.1)	3.3%	(35.6, 266)	5.3%	1
	(100, 150)	(101.2, 146.8)	1.07%	(92.9, 149.8)	2.36%	2

**Table 2 sensors-23-02050-t002:** The locations of the through-hole in the diamond-shaped arrays with different sizes (Unit: mm).

Number	Position of Array	Center Position of the Hole	Localization Result (Simulation)	Error	Localization Result (Experiment)	Error
3	Diagonal distance was 80, A (150, 110), B (110, 150), C (150, 190), D (190, 150)	(230, 230)	(211.1, 211.1) (211.3, 211.3)	6.3% 6.2%	(244.6, 253.1)	7.7%
4	Diagonal distance was 120, A (150, 90), B (90, 150), C (150, 210), D (210, 150)		(218.6, 220.7)	3.8%	(251.9, 249.1)	7.3%
5	Diagonal distance was 160, A (150, 70), B (70, 150), C (150, 230), D (230, 150)		(220.4, 219.8)	3.4%	(244.1, 243.1)	4.7%
6	Diagonal distance was 200, A (150, 50), B (50, 150), C (150, 250), D (250, 150)		(221.6, 223.4)	2.8%	(241.4, 239.3)	3.8%

## Data Availability

The data presented in this study are available on request from the corresponding author.

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
