# Peer review of "A Novel Baseline-Free Method for Damage Localization Using Guided Waves Based on Hyperbola Imaging Algorithm"

_sensors, 2023, doi:10.3390/s23042050_

Round 1

Reviewer 1 Report

Interesting paper. I have the following comments:

The sentences in lines 130 and 133 are confused. Authors talk about an “intact state”. Authors should provide a better explaining of this concept in order to define the foundations of the new method they are commenting.

At lines 150-151, authors state that “Hilbert transform is used to estimate TOF” but it is not clear how they extract the time from the envelope of the signal component, neither they provide a reference to complete this information.

Integral operators in expressions 6.1 and 6.2 exhibit different sizes. These sizes should be normalized.

In line 165 says: “has be determined”. It should be corrected as: “has been determined”.

In line 212 says “Entity unit C3D8R”. What is this entity? It is not clear.

Line 231: It says thar “The envelope curve … was constructed by Hilbert transform (Eq (5))…”. It is not clear how this curve was constructed. Neither an adequate reference has been provided to complete the explanation.

Lines 235-237: It is not obvious how the group velocity of 5164.7m/s for S0 mode was calculated. This issue should be clarified.

Line 287: What do the authors want to say with “256-time averaging”? 256 sample-time averaging?

Lines 290-291: This sentence is confused. The expression “intact state” and its context should be clarified.

Line 295: Again, how is the group velocity for S0 mode was computed in this experimental stage?

Lines 310-312: How are phase compensation and standard normalization performed? Results are provided but there is no an explanation neither a reference to complete this information.

Lines 318, 392, 395, 397: The way that these titles refer to imagens is confused. It is more clear to put “:” before “a” and “;” before “b”.

Line 317: Fonts in horizontal labels at Figure 7 are too small. The scale “x10“ is unreadable.

When the paper was printed, Figure 11 was split; so did Table 2. Be sure of the integrity of that figures and tables.

Reviewer 2 Report

This work aims to theoretically, numerically, and experimentally present a baseline-free method for damage localization using guided waves based on the hyperbola imaging algorithm. In my opinion, the paper in question needs revision before it can be considered for publication in the Sensors journal. Some things to think about are as follows:

1-      The primary aim of this study is to achieve damage localization based on the hyperbola imaging algorithm with fewer transducers. The novelty introduced lies within the diamond or the square form of array of transducers. Although the introduced arrangement was successful for damage localization, this work can be improved further by comparing the accuracy of the proposed arrangement to the traditional hyperbolic arrangement presented in the literature.

2-      The authors presented a study of different sizes of diamond shaped arrays, where limitations of the diagonal distance were estimated. However, due to the central layout of the proposed array within the structure, the boundary reflections were eliminated. Thus, the authors should also present the limitations for the near boundary layout of the proposed array.   

3-      The references should be updated with the most recent references. Notably, only one of the thirty references was made in the year 2020. There are no references in the following years.

Reviewer 3 Report

This paper is focused on SHM based on novel method using Lamb wave, the topic is very interesting. Symmetrical sensor arrays were adopted to eliminate direct wave, and cross correlation method was used to determine the time difference. Numerical and lab experiments verified the proposed method. And the paper is well designed with good writing. Therefore, the paper is suitable for publication with minor revised.

(1)  Improved in figures for easy understanding, i.e., four points in Fig.1 should marked with A, B, C, D. The sign ‘+” should be improved with other color for Fig.9 and Fig.10,

(2)  Only 1mm thickness of specimen was tested, please clarify whether the thickness affect the proposed method.

Reviewer 4 Report

Authors tend to present a novel baseline-free method for damage localization in this manuscript and comments are listed as follows.

1 It is suggested that motivation should also be included in the part of abstract.

2 The contribution of this manuscript should be strengthened in the part of introduction.

3 In the title of this manuscript, the proposed method is based on hyperbola imaging algorithm. But in the 2nd part of this manuscript, description of hyperbola imaging algorithm is not presented.

4 Authors should present a flowchart of the proposed method.

5 In the part of experiment lacks comparison.

6 Authors should cite more references which are published within 5 years.

Round 2

Reviewer 2 Report

The authors are sincerely addressed all the mentioned points.

Author Response

Thank you very much for your suggestions on our manuscript.

Reviewer 4 Report

1 The flowchart in figure 2 is not proper. Authors should read other papers and learn how to present a flowchart. Content in flowchart should be concise.

2 It is common to present comparison results to prove the validity of the proposed method. It is ridiculous that authors can not understand the meaning of lacking comparison. It is noticed that authors cite lots of references about damage localization. Authors could choose several methods in references for obtaining comparison results in this manuscript. At least one of comparison methods should be proposed within 5 years.

Round 3

Reviewer 4 Report

After revision, the current version is OK for Sensors.